# Bond selectivity in electron-induced reaction due to directed recoil on an anisotropic substrate

Kelvin Anggara[1], Kai Huang[1], Lydie Leung[1], Avisek Chatterjee[1], Fang Cheng[1,†] & John C. Polanyi[1]

Bond-selective reaction is central to heterogeneous catalysis. In heterogeneous catalysis, selectivity is found to depend on the chemical nature and morphology of the substrate. Here, however, we show a high degree of bond selectivity dependent only on adsorbate bond alignment. The system studied is the electron-induced reaction of meta-diiodobenzene physisorbed on Cu(110). Of the adsorbate's C-I bonds, C-I aligned 'Along' the copper row dissociates in 99.3% of the cases giving surface reaction, whereas C-I bond aligned 'Across' the rows dissociates in only 0.7% of the cases. A two-electronic-state molecular dynamics model attributes reaction to an initial transition to a repulsive state of an Along C-I, followed by directed recoil of C towards a Cu atom of the same row, forming C-Cu. A similar impulse on an Across C-I gives directed C that, moving across rows, does not encounter a Cu atom and hence exhibits markedly less reaction.

[1] Lash Miller Chemical Laboratories, Department of Chemistry and Institute of Optical Sciences, University of Toronto, 80 St George Street, Toronto, Ontario M5S 3H6, Canada. † Present address: National University of Singapore, Center for Advanced 2D Materials, 1 CREATE Way, CREATE Tower, Singapore 138602, Singapore. Correspondence and requests for materials should be addressed to J.C.P. (email: jpolanyi@chem.utoronto.ca).

Bond-selective chemistry due to differences in energy barriers continues to play a major part in the understanding of thermal heterogeneous catalysis[1,2]. A notable advance in recent years has been the employment of selective excitation of a molecule incident or present on a surface as a means to influence the reaction probability. For vibrationally and translationally excited gaseous beams incident at metal surfaces, reactions have been reported to be mode-specific[3–5], bond-selective[6,7] and gas-phase alignment-dependent[8]. In the case of molecules adsorbed at surfaces, electrons from scanning tunnelling microscopy (STM) have been employed to excite vibrationally or electronically the adsorbate, resulting in molecular translation[9–11], rotation[11–13], isomerization[14] and desorption[9,15,16]. The breaking of a selected chemical bond in the adsorbate has also been achieved by varying the electron energy[16–19] or, for extended adsorbates, the excitation location[19].

Here, we show a high degree of bond selectivity due to a cause that, to our knowledge, has not previously been noted, namely the directed recoil of products formed as a result of electron-induced surface reaction. The approach is quite general since directed recoil of products along what was the prior bond direction has been widely reported[16,20–27]. When the directed recoil occurs at an anisotropic surface, we find bond-selective surface reaction occurs. We show that different alignments, even of a chemically identical bond, can result in a hundred-fold alteration in reaction probability, with corresponding bond selectivity. The example given is that of an electron from an STM tip impinging on a single meta-diiodobenzene (mDIB) molecule on Cu(110) giving two orders of magnitude greater probability of breaking a carbon–iodine (C-I) bond lying 'Along' (AL) a Cu row than the C-I that lies 'Across' (AC) the rows. Molecular dynamics (MD) calculations employing an approximate anionic potential, followed by impulsive recoil across a ground potential computed by density functional theory (DFT) reproduce the observed bond selectivity, explaining it as due to directed recoil across the anisotropic surface of the Cu.

## Results

**Experiment.** Experiments were performed on a Cu surface held at 4.6 K. Two physisorbed states of mDIB were observed: one termed 'Row'[28] and the other 'Diagonal', the latter being the principal subject of the present report. An STM image of the Diagonal adsorption state on Cu(110) is shown in Fig. 1a, with a theoretical simulation directly below.

From the simulation, the benzene ring is located at a short-bridge site with one C-I directed along the Cu row and the other at 126° from the Cu row (measured clockwise from [1$\bar{1}$0]), leading to the two lobes that were observed, corresponding to the two C-I bonds with I atoms located atop Cu atoms, as evident in the simulated STM image. The bond lengths of both C-I bonds in the physisorbed state were computed as ~2.1 Å, and the heat of adsorption as 1.0 eV. This Diagonal adsorption state was also predicted by a computational study performed by Panosetti and Hofer[29]. The midpoint between the two lobes in Fig. 1a, EXPT, is indicated by a white cross used as the origin of the spatial distribution of the products following electron-induced reaction; see Fig. 1a–c.

Electron-induced reaction of physisorbed mDIB was initiated by tunnelling electrons from the STM tip placed over the centre of the intact physisorbed reagent at a sample bias ≥ + 1.0 V. The experimental findings are illustrated in Fig. 1, with, at left, sample STM images for the initial and final states of the major reaction path breaking the C-I bond AL the Cu rows, and, at the right, the minor path breaking the C-I bond directed AC the Cu rows. For 139 reactive cases, the breaking of only one of the two C-I bonds

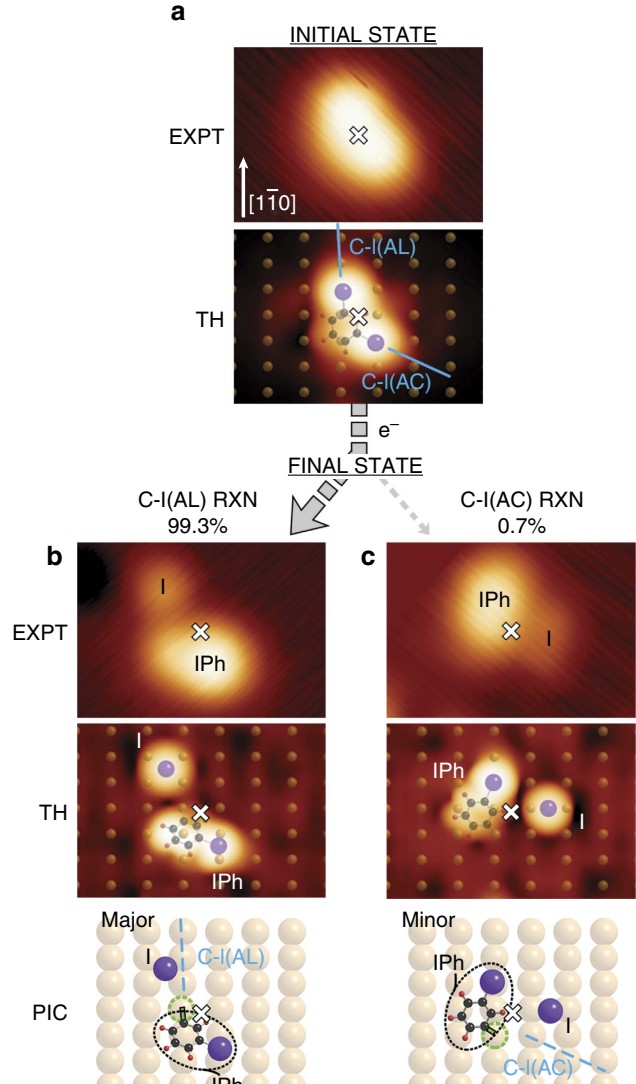

**Figure 1 | Bond-selective electron-induced reaction of physisorbed mDIB on Cu(110).** STM images (EXPT) and simulations (TH) show the selective dissociation of the C-I bond. (**a**) An intact physisorbed mDIB. The white cross indicates the midpoint of mDIB in the STM image. The two C-I bond directions, 'AL' and 'AC', are given by blue lines and designated as C-I(AL) and C-I(AC). (**b**) A reactive outcome as a result of the breaking of C-I(AL) in the prior mDIB reagent. (**c**) A reactive outcome for the single observed case of mDIB breaking C-I(AC). In the pictured final states (PIC), the green dashed circle indicates the Cu atom to which the IPh was bound, and the blue dashed line the direction of the broken C-I bond. All experimental STM images shown here are 25 × 17 Å$^2$ in size, recorded at a sample bias of + 0.1 V and a tunnelling current of 0.1 nA.

was observed, namely, the AL bond of the physisorbed mDIB, in preference to the AC bond. The bond selectivity amounted to a 99.3% probability for C-I(AL), as against 0.7% for the alternate C-I(AC).

A representative case of C-I(AL) bond breaking is shown in Fig. 1b. The reaction products are a chemisorbed I atom and a chemisorbed iodophenyl (IPh). A comparison between experiment (Fig. 1b, EXPT) and theory (Fig. 1b, TH and PIC) allowed us to identify the binding sites of both products with respect to the prior mDIB. The I atom was found at the closest four-fold hollow site adjacent to the initial I atom position in the reagent. The IPh product was bound to the atop Cu atom underneath the

reagent mDIB (green dashed circle, Fig. 1b, PIC). The I atom and IPh were found to recoil in opposite directions AL the Cu row (Supplementary Fig. 1a and Supplementary Note 1), as expected for the dissociation of C-I(AL) aligned along the row. The most probable recoil distance of the I atom measured from the white cross of Fig. 1 was 4.7 Å. Figure 1c and Supplementary Fig. 1b show the minor path consisting of 1 out of 139 cases. In this case, the products were found to recoil in the opposite direction AC the rows (Supplementary Note 1), from which we conclude that the bond dissociated was C-I(AC).

The observed strong preference for the breaking of C-I(AL) rather than C-I(AC) shows that the electron-induced reaction of the Diagonal physisorbed mDIB is markedly bond-selective. As the reaction of Row physisorbed mDIB did not exhibit any detectable bond selectivity between its symmetrical C-I bonds[28], the bond-selective reaction of Diagonal is attributed to its asymmetric adsorption geometry. This reactive bond selectivity will be examined below by the MD theory.

The number of electrons involved in triggering a reactive event was determined experimentally for the major path by measuring the average reaction rate as a function of the tunnelling current in the range of 0.6–18 nA, at a constant sample bias of +1.3 V. As shown in Fig. 2a, the reaction rate scaled linearly with the tunnelling current, evidencing a one-electron process. This linear relationship also excludes an electric field effect as a major cause of reaction[30].

As in previous studies of the electron-induced reaction of aryl iodides by ∼1 eV electrons on Cu(110)[26–28,31,32], we ascribe the reaction to adsorbate electronic excitation. The computed projected density of states is shown in Fig. 2b. It exhibits a nodal plane between carbon and iodine atoms for the lowest unoccupied molecular orbital (LUMO), indicative of a σ* antibonding character with respect to the C-I bond. This C-I antibonding orbital was computed to be 0.8 eV above the Fermi level. The yield for electrons of ∼1.3 eV was measured as ∼$10^{-9}$ reactive events per electron for the major path. The high single-electron energy of at least 1.0 eV required to give reaction argues against vibrational excitation as the source of induced reaction.

**Theory.** The observed bond selectivity can be understood in terms of the different recoil directions of the AL and AC fragments following an electron-induced repulsive impulse in the LUMO of one or the other C-I bond, leading to different reaction paths across the anisotropic Cu(110) surface. As in previous work[25–28,31,32] we used the 'Impulsive Two-State' (I2S) model to simulate the MD of the electron-induced reaction. In this two-electronic state model, the MD was first followed for the 192-atom system on an approximate anionic repulsive potential-energy surface (PES) obtained by the transfer of an electron to the valence shell of a halogen atom for a period of femtoseconds (a time $t^*$). Thereafter, the atoms were returned with their accumulated momenta to the *ab initio* DFT ground PES for MD over a period of picoseconds needed to reach the reacted final states. In the present case, the added charge comprises one electron at the I atom of C-I(AL) to favour breaking of that bond, or alternatively one electron at the I atom of C-I(AC) to favour the alternate bond breaking.

Figure 3a shows the dynamics for an electron added at C-I(AL), using the minimum $t^*$ of 20 fs required for reaction. Figure 3a shows that the impulse, with momenta carried over to the ground PES, stretched C-I(AL) causing it to break at ∼150 fs concurrently with the formation of the C-Cu (i.e., IPh-Cu) and I-Cu bonds. The distance versus time plot of Fig. 3a' in the panel below shows bond extension of C-I(AL) from its initial separation of 2.1 to a 3.0 Å, accompanied by the formation of C-Cu and I-Cu

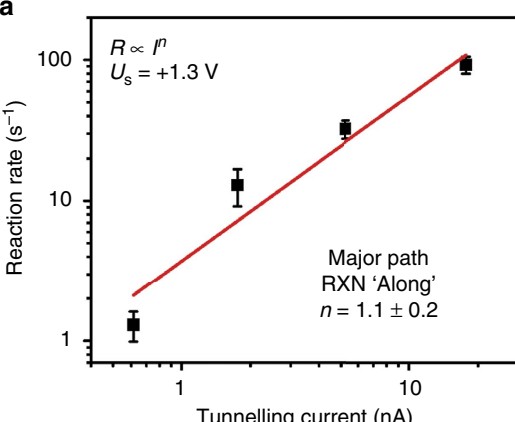

a

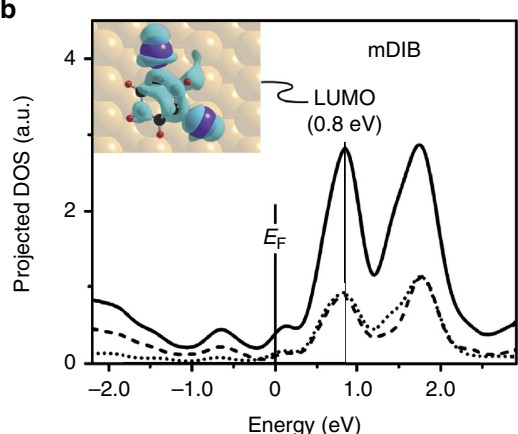

b

**Figure 2 | Evidence of electron-induced process.** (**a**) Rate versus current plot (in log scale) for the electron-induced reaction of physisorbed mDIB on Cu(110). The plot is shown for C-I(AL) reaction (major path) with a linear fit (red line) of the data. The slope of the linear fit was determined to be 1.1 ± 0.2. The good quality of the fit was evidenced by the coefficient of determination ($R^2$) of 0.927. Reaction rates were measured at +1.3 V. The error bar is the standard error from the exponential fitting of the data at each current (see Methods for details). (**b**) Projected-density-of-states (pDOS) calculation for the physisorbed mDIB. The pDOS shows the LUMO of mDIB to be 0.8 eV above the Fermi level. The pDOS of the mDIB molecule is given by the black line; the dashed and dotted lines give the pDOS of the I atoms and the C atoms next to the I atoms. The inset in panel **b** visualizes the LUMO of mDIB (isocontour = 0.0005 e Å$^{-3}$). The nodal planes between the C and I atoms show the LUMO to be of σ*(C-I) antibonding character.

bonds at their equilibrium separations (all equilibrium separations being indicated by horizontal dotted lines). Figure 3b shows that for the same $t^*$ the placing of one electron in the I atom of the alternate C-I(AC) bond did not break that C-I bond. This failure to break C-I(AC), as evident from the distance versus time plot of Fig. 3b', is connected with the larger separation between the C atom of C-I(AC) from its nearest Cu atom neighbour (C-Cu is 2.3 Å in the AL case, 3.0 Å for AC). We show below that this difference in C-Cu separation is significant for the extent of C-Cu binding energy.

In Fig. 4, we superimpose the MD taken from a full calculation of the motion of the 192 atoms onto a two-dimensional cut through the potential-energy hypersurface that gives the dependence of the potential energy on the two C-I separations. In drawing this PES we have kept the angle between the two C-I

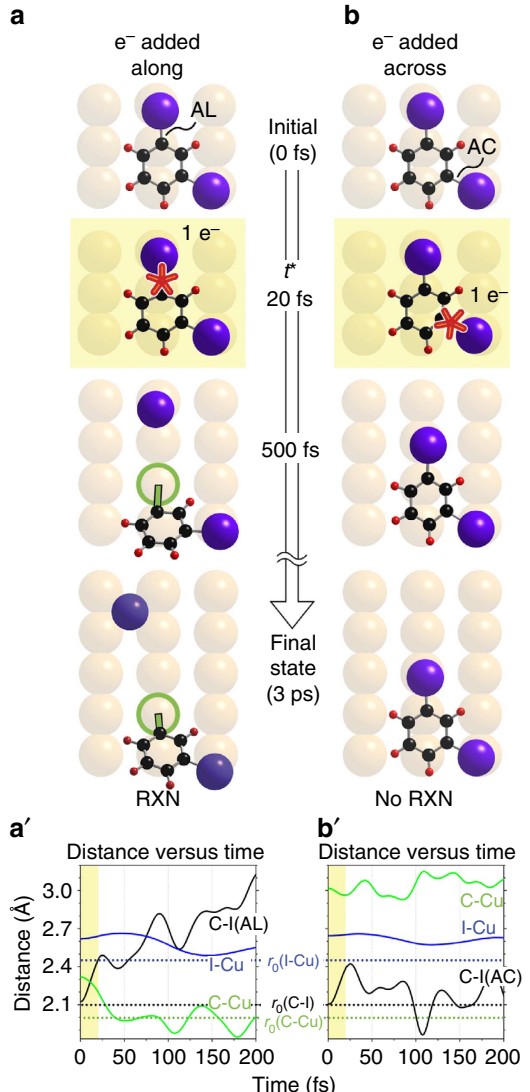

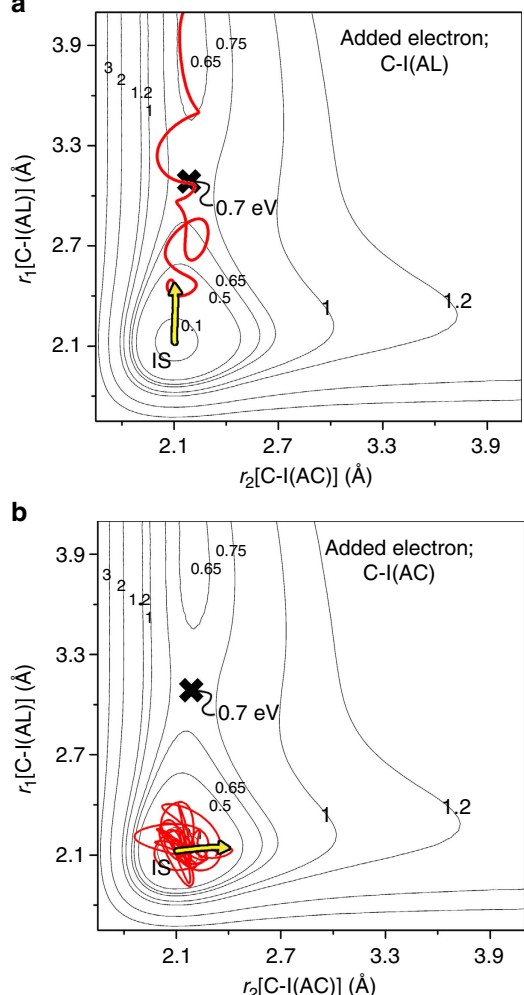

**Figure 3 | Computed trajectories for the bond-selective reaction of mDIB using the system's 192 atoms.** The trajectories, calculated using the I2S model with $t^* = 20$ fs, give the time evolution of the system to its final state. (**a**) Trajectory for one electron placed at the I atom of C-I(AL); the green circle indicates the nearest Cu atom to the C atom of C-I(AL). (**b**) Trajectory for one electron placed at the I atom of C-I(AC). The yellow highlights indicate that the system is in the anionic state. In (**a′**) and (**b′**) below, the time evolution of the C-I, I-Cu and C-Cu separations are shown as black, blue and green lines, respectively, for each trajectory, AL and AC. The C-Cu bond separation is measured from the Cu atom nearest to the carbon of the C-I bond: C-I(AL) in (**a′**) and C-I(AC) in (**b′**). The horizontal dotted lines give the equilibrium separations of the atomic pairs, as indicated. I2S, Impulsive Two-State.

**Figure 4 | Trajectories using the I2S model superimposed on a restricted ground PES.** The restricted cut through the ground potential-energy hypersurface was obtained by varying only the two C-I bond distances at a fixed bond angle of 120°. The trajectory from the I2S model was based on a full molecular dynamics calculation using all 192 atoms in the system. (**a**) The trajectory for one electron added to the I atom of the AL bond, C-I(AL). (**b**) The trajectory for one electron added to the I atom of the AC bond, C-I(AC). The energy values shown are relative to the initial state (IS). The yellow arrow in the trajectories encompass the 20 fs duration of the repulsion on the anionic PES; the red line indicates the subsequent ∼3 ps motion on the ground PES. I2S, Impulsive Two-State.

bonds fixed at its initial 120°. This constant angle is a good approximation to the PES governing the initial reaction dynamics, since the motion calculated for the full hypersurface shows the angle between the two C-I bonds decreasing by only 10° in the first ∼200 fs, during which C-I(AL) extends by 1.0 Å. On this restricted PES, the energy barrier to dissociate C-I(AL) is shown to be 0.7 eV for that coordinate, while the barrier to dissociate C-I(AC) is markedly higher for the alternate bond extension (the crest of the barrier along C-I(AC) is at 1.5 eV, lying beyond the range of Fig. 4).

We attribute the lower barrier along C-I(AL) to the greater C-Cu binding, and hence greater stabilization along this coordinate than along C-I(AC) (see Fig. 3a′,b′ which show the C-Cu bond forming AL, but not AC). The extent of stabilization due to C-Cu binding can be estimated from the Bader version of the bond-length versus bond-order ($n$) relationship for atomic pairs[33], which gives $n = 0.5$ for C-Cu at the transition state (C-I(AL) = 3.0 Å) in Fig. 4a as compared with $n = 0.3$ for extension of the other coordinate, C-I(AC), to 3.0 Å. For a computed ∼3.0 eV bond energy of C-Cu, this leads to ∼0.6 eV greater stabilization due to C-Cu binding in the AL coordinate than the AC coordinate. Accordingly, the lower barrier for the C-I(AL) (0.7 eV) than C-I(AC) (>1.2 eV) in Fig. 4 can be ascribed to stronger C-Cu binding and hence greater stabilization AL rather than AC. This finding shows that the binding of the products to the surface, late in the course of reaction, can influence reaction barriers with resultant bond selectivity.

The trajectories shown in Fig. 4 are taken from the motion on the full potential, with the two C-I bond separations plotted here on the 'fixed bond-angle' PES whose energy contours are shown. Bond selectivity is caused by the impulsive motion on the anionic PES, which impels the system on the ground PES to stretch C-I(AL) giving surface reaction over a 0.7 eV barrier along $r_1$, but does not stretch C-I(AC) sufficiently to surmount the higher barrier along $r_2$. The low barrier for C-I(AL), as indicated above, is due to the proximity of an underlying Cu atom along the row direction, stabilizing the system through C-Cu bond formation.

It should be noted that the bond selectivity being reported is specific to the impulsive dynamics (see Fig. 4) being a consequence of the directed recoil of the atoms liberated in electron-induced reaction. The C-I bond breaking could also in principle be obtained thermally, although this was not studied here. We have, however, calculated, ab initio, the different energy barriers to reaction for the individual breaking of the two C-I bonds by C-I extension along the alternative minimum-energy paths of the ground PES. The computed barrier heights relevant to thermal reaction were found to be 0.3 eV for the previously favoured AL C-I and 0.2 eV for the previously disfavoured AC C-I. Thermal reaction is therefore predicted to exhibit a modest bond selectivity but in the opposite sense to that observed and modelled here for electron-induced reaction.

## Discussion

The high degree of bond selectivity in electron-induced surface reaction reported here is linked to directed recoil in relation to the anisotropy of the substrate, resulting in stronger bonding to the substrate when one bond is broken than the other. We anticipate similar bond-selective reaction for electron-induced processes on other anisotropic surfaces, the cause being the greater proximity of a substrate atom to one bond in an adsorbate than to another. This proximity leads in the present instance to $100 \times$ greater reactivity for the adsorbate bond more closely adjacent to a substrate atom, despite the fact that the bonds being broken in the reaction are chemically identical. The proximity, in turn, is determined by the geometry of the physisorbed adsorbate relative to the substrate atoms. Directed recoil together with substrate anisotropy can therefore be expected to lead to bond selectivity in other electron-induced surface reactions, for example, in chemical reactions at surfaces being subjected to generalized electron radiation, in an extension of conventional heterogeneous catalysis.

## Methods

**Experiment.** The experiments were conducted at 4.6 K, using an ultrahigh vacuum Omicron low-temperature STM with a base pressure of $< 3.0 \times 10^{-11}$ mbar. The Cu(110) surface was prepared by cycles of $Ar^+$ sputtering (0.6 keV) and annealing (800 K). mDIB (98%; Sigma-Aldrich) was outgassed by 3–4 cycles of freeze pump thaw before being dosed via a capillary tube directed at the Cu surface. During dosing the substrate temperature rose to 12.6 K.

The surface was imaged by STM using the constant-current mode. The bias reported was the sample bias. The reactions of mDIB were induced by tunnelling electrons from the STM tip. This is briefly described as follows: (i) after imaging an intact physisorbed mDIB molecule, the STM tip was placed over the molecule (white cross in Fig. 1); (ii) the tip height was adjusted according to a predefined set of sample bias ($V_{set}$) and tunnelling current ($I_{set}$); (iii) the feedback loop was deactivated, and a surface bias ($V_{reaction}$) was applied for up to 10 s while the tunnelling current ($I_{reaction}$) was recorded as a function of time. The reaction was identified by a single discontinuity of the current at time $t$. Subsequent imaging of the same area confirmed the occurrence of the electron-induced reaction products. Their locations were obtained relative to the reagent position, using the WSxM software[34].

The reaction order with respect to the current was established by examining the reaction rate as a function of the current ($I_{reaction}$). To determine the reaction rate ($R$), we have followed the procedure first described by Ho and co-workers[12] and modified it for two competing pathways[28]. The times ($t$, see above) measured from repeated experiments at a particular $I_{reaction}$ were binned according to the Doane's formula[35]; the resulting histogram being normalized and fitted with a single-

parameter exponential function ($e^{-Rt}$). The value of $R$ obtained from the exponential fitting was plotted for each $I_{reaction}$ as a log–log graph with an error bar that represents the standard error from the exponential fitting. The slope of the linear fit between $R$ and $I_{reaction}$ in the log–log graph gives the reaction order with respect to the current. In the linear fitting, each point was weighted by its standard error.

**Theory.** Plane-wave-based DFT calculations were performed using Vienna Ab Initio Simulation Package (VASP 5.2.11)[36,37] installed at the SciNet supercomputer[38]. The parameterization followed the earlier work[26–28,31,32] of this laboratory, using the projected augmented wave, generalized-gradient approximation[39,40], Perdew–Bruke–Ernzerhof (PBE) functional[41] and the second version of Grimme's semiempirical correction for dispersion force corrections (PBE-D2)[42]. Calculations of the geometry relaxation, MDs and barrier height were restricted to the Γ-point at the centre of the surface Brillouin zone, using a cutoff energy of up to 450 eV with the dipole correction along the surface normal direction and no spin polarization. The Cu(110) surface was represented by a (6x6) slab that consisted of 180 Cu atoms in five layers with a vacuum gap of 16 Å. In the geometry relaxation calculations, all atoms were allowed to relax, except the bottom two layers of the Cu-slab, until the residual force on each atom was $< 0.01$ eV Å$^{-1}$. Given that PBE-D2 overestimates the physisorption energy[43], the energy was corrected by RPBE-D2 in a denser k-mesh (3x3x1) from the structure relaxed using PBE-D2. The STM simulation used the Tersoff-Hamann approximation and was visualized using the Hive software[44,45]. The molecular structures and the electron-charge densities were visualized using the VESTA software[46].

The Impulsive Two-State model[25–28,31,32] implemented in the MD calculations was applied to simulate the electron-induced reaction. To introduce the C-I repulsion caused by an electron occupying the LUMO of the intact mDIB, we used the anionic pseudopotential method[47,48], which has been developed and tested in a number of cases in this laboratory for electron-induced reaction at both metallic[25–28,31,32] and semiconducting surfaces[49]. In this method, an electron was placed in an I atom of the molecule by exciting the 4d core electron of the I atom to its 5p valence shell. The MD calculations were run as a microcanonical ensemble using a time step of 0.5 fs. The reaction trajectory was obtained by evolving the system on the anionic PES for a brief residence time of $t^*$, and afterwards, on the ground PES for up to ~3 ps. The resulting product distribution was compared with the experimental one.

The Climbing-Image Nudged Elastic Band technique[50] was used to compute the minimum-energy path and to locate the transition states for mDIB dissociation on Cu(110) on the ab initio ground PES. The Climbing-Image Nudged Elastic Band calculations used the initial and final states observed in the electron-induced experiment. The number of images used, including the initial and final states, was 10 for C-I(AL) and 13 for C-I(AC). The calculations were conducted until the forces orthogonal from the band were $< 0.02$ eV Å$^{-1}$.

**Data availability.** The authors declare that the data supporting the findings of this study are available from the corresponding author on request.

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

## Acknowledgements

We thank Oliver MacLean and Dr Zhixin Hu for helpful discussions, and Dr Zhanyu Ning and Chen-Guang Wang for assistance with the DFT calculations. This work was funded by the Natural Sciences and Engineering Research Council of Canada (NSERC) and the University of Toronto NSERC General Research Fund. Computations were performed on the Tightly Coupled System (TCS) at SciNet HPC Consortium. SciNet is funded by the Canada Foundation for Innovation under the auspices of Compute Canada, the Government of Ontario, Ontario Research Fund–Research Excellence and the University of Toronto. K.A. thanks the Connaught International Scholarship for Doctoral Students for financial support.

## Author contributions

K.A., K.H. and L.L. contributed equally to this work. J.C.P. designed and supervised the project. K.A., K.H., L.L., A.C. and F.C. collected the STM data. K.A., K.H., L.L. and A.C. analysed the STM data. K.A. conducted the *ab initio* calculations. All the authors contributed to the manuscript.

## Additional information

**Competing financial interests:** The authors declare no competing financial interests.

**Publisher's note**: 

