## [Peer Review File · Nature Communications]

Reviewers' comments:

Reviewer #1 (Remarks to the Author):

A) The authors are to be commended on a carefully done experimental study supported by electronic structure and molecular dynamics calculations. The work uses an STM to study the C-I cleavage reaction of meta-di-iodo benzene on the Cu(110) surface. Tunneling electrons from the STM activate bond cleavage, and STM imaging before and after the reaction reveal the initial and final location of reagents and reaction products, respectively. The key result reported is that the dissociation probability for C-I bonds aligned along the rows of Cu atoms on the surface are nearly 100-fold more likely to dissociation.

B) The experiments provide a nice visual example of how surface structure can lead to selective bond cleavage in this reaction, and the accompanying calculations provide a plausible explanation of the effect. The experimental and computational methodology follows very closely other studies from the research group, with the studies described in Refs. 26, 27, and 28 being most closely related. The most novel finding in this paper is the observation of the large reactivity difference for C-I bonds oriented along and across the Cu rows.

The computational work suggests that the origin of this effect is the location of Cu atoms on the surface, and their proximity to the potential reaction products. In the case of C-I across the rows, the Cu atoms needed to bond the C and I fragments are not optimally located, and there is consequently a higher energetic barrier associated with dissociating that bond. The reported difference in barrier height of 0.5 eV or more easily explains the differential reactivity and high selectivity for along-row C-I bond cleavage that the authors observed.

While the experiments provide a beautiful visual example of this effect, the finding that reaction barriers for the two reaction channels differ by such a great amount makes the observation of bond selective reactivity, and even the dramatic extent of this selectivity, less than surprising. The use of selective coordination or stabilization of a transition state to modulate the energetic barriers for competing reaction paths is a long-standing tool of synthetic chemists, and even on surfaces, it is well established that different reagent binding sites (e.g. a flat terrace vs. a step or kink site) can lead to different reaction patterns.

Therefore, I find the originality and novelty of this work to mostly lie in the experimental realization of a very visual illustration of this well understood chemical effect.

C. Overall, the data appear to be of good quality and are clearly presented and described.

My only question here would lie with the interpretation of Fig. 2a. The error bars, as illustrated, do not provide compelling support for the linear dependence claimed. In fact the first two points appear to have a slope closer to two, while the second through fourth points do indeed appear to be linear. The best-fit line barely overlaps with any of the error bars. It would be helpful if the authors clearly stated the confidence limit shown by the error bars and commented on this suitability of a linear fit to the data.

D. See comment in C. above. Otherwise fine

E. The authors' conclusions appear to be supported by the work. I do have a suggestion, though.

I felt as though the authors' strong emphasis on achieving bond selectivity in the abstract and conclusion may have deflected attention from what I see as the more novel and significant results of this work that relate to reaction mechanism. As noted above, the fact that the reaction barriers for the two reaction paths differ by half an eV or more essentially guarantees a very high degree of bond selectivity in favor of the reaction channel with the lowest barrier. This is not a surprising

result (it is essentially the basis of all of modern synthetic chemistry).

Instead, I feel the more interesting and impactful result is how prompt, electron stimulated reactions can lead to directed product recoil, and how the recoil process, when coupled with surface anisotropy, impacts the energy landscape between reactants and products. These ideas do appear in the paper, but in my reading of the paper, I found the connection between these ideas was obscured by the emphasis on bond selectivity.

Stated another way, it seems to me that the authors are using the prompt dissociation event to access different initial reaction paths/ trajectories for the two orientations of C-I bonds. The energy landscape along those two paths differs significantly due to the anisotropy of the surface, and this leads to the observed bond selectivity. This effect is illustrated very nicely in Fig. 4. Perhaps connecting the discussion and conclusion more closely to this illustration would help convey this point to readers.

Second, the work emphasizes how important product binding can be in dictating reaction selectivity. Surface science has many examples of where differences in reagent binding (physisorption vs. chemisorption, or different chemisorption interactions) alter reactivity patterns, but this experiment provides a nice example of how product binding alone can influence reactivity.

So I'd encourage the authors to consider highlighting these other points in the paper.

F. I don't see any need for additional experiments or calculations. See above for suggested improvements to the manuscript.

Also, on p. 4, the authors refer to the pDOS for the molecular as appearing in Fig. 3b. I believe it's in Fig. 2b.

G. References seem fine.

H. Clarity - see above comments above regarding the emphasis on bond selectivity.

Reviewer #2 (Remarks to the Author):

In the manuscript by Anggara et al., a selective bond dissociation of an adsorbate molecule is described that was investigated by using low-temperature STM. The dissociation of the C-I bond in meta-diiodobenzene is induced by the injection of a tunneling electron that is attached to the molecular state (LUMO). Most importantly, they found that the dissociation of one of the C-I bonds is strongly preferred compared to another because of the C-I bond orientation with respect to the anisotropic atomic arrangement of the Cu(110) surface. The experimental finding is supported by their theoretical calculations. The reported result is one of the clear examples that demonstrate the importance of atomic-scale environment on a chemical reaction at a metal surface. Particularly, the observed bond-selectivity is very difficult to identify with the other spectroscopic methods that observe a signal from molecular ensemble. I recommend the manuscript to be published in Nature communications.

I have a few comments;

1. Does the dissociation never occur at the opposite bias polarity? What happens when a hot hole is attached to the molecule?
2. How is the "average rate" and its error in Fig. 1a determined?
3. Is there any specific and general example that shows the importance of the selective bond dissociation presented in the manuscript? How is the knowledge obtained from single-molecule experiments useful to understand the process in a heterogeneous catalysis?

4. Is the observed bond-selectivity unique for an electron-induced process? Is this also expected for a thermally-induced process?

Reviewer #1:

A) *The authors are to be commended on a carefully done experimental study supported by electronic structure and molecular dynamics calculations. The work uses an STM to study the C-I cleavage reaction of meta-di-iodo benzene on the Cu(110) surface. Tunneling electrons from the STM activate bond cleavage, and STM imaging before and after the reaction reveal the initial and final location of reagents and reaction products, respectively. The key result reported is that the dissociation probability for C-I bonds aligned along the rows of Cu atoms on the surface are nearly 100-fold more likely to dissociation.*

Response to (A): No question from Reviewer #1.

B) *The experiments provide a nice visual example of how surface structure can lead to selective bond cleavage in this reaction, and the accompanying calculations provide a plausible explanation of the effect. The experimental and computational methodology follows very closely other studies from the research group, with the studies described in Refs. 26, 27, and 28 being most closely related. The most novel finding in this paper is the observation of the large reactivity difference for C-I bonds oriented along and across the Cu rows.*

The computational work suggests that the origin of this effect is the location of Cu atoms on the surface, and their proximity to the potential reaction products. In the case of C-I across the rows, the Cu atoms needed to bond the C and I fragments are not optimally located, and there is consequently a higher energetic barrier associated with dissociating that bond. The reported difference in barrier height of 0.5 eV or more easily explains the differential reactivity and high selectivity for along-row C-I bond cleavage that the authors observed.

While the experiments provide a beautiful visual example of this effect, the finding that reaction barriers for the two reaction channels differ by such a great amount makes the observation of bond selective reactivity, and even the dramatic extent of this selectivity, less than surprising. The use of selective coordination or stabilization of a transition state to modulate the energetic barriers for competing reaction paths is a long-standing tool of synthetic chemists, and even on surfaces, it is well established that different reagent binding sites (e.g. a flat terrace vs. a step or kink site) can lead to different reaction patterns.

Therefore, I find the originality and novelty of this work to mostly lie in the experimental realization of a very visual illustration of this well understood chemical effect.

Response to (B): No question from Reviewer #1.

C) Overall, the data appear to be of good quality and are clearly presented and described.

My only question here would lie with the interpretation of Fig. 2a. The error bars, as illustrated, do not provide compelling support for the linear dependence claimed. In fact the first two points appear to have a slope closer to two, while the second through fourth points do indeed appear to be linear. The best-fit line barely overlaps with any of the error bars. It would be helpful if the authors clearly stated the confidence limit shown by the error bars and commented on this suitability of a linear fit to the data.

Response to (C): The authors have now addressed this important question in the Methods (highlighted in red): “The value of R obtained from the exponential fitting was plotted for each current as a log-log graph with an error bar that represents the standard error from the exponential fitting.”

To address the suitability of the linear fit to the data, we have now stated in the captions of Fig. 2 (highlighted in red): “The good quality of the linear fit is evidenced by the coefficient-of-determination (R^2) of 0.927.”

D) See comment in C. above. Otherwise fine

Response to (D): No question from Reviewer #1.

E) The authors' conclusions appear to be supported by the work. I do have a suggestion, though.

I felt as though the authors' strong emphasis on achieving bond selectivity in the abstract and conclusion may have deflected attention from what I see as the more novel and significant results of this work that relate to reaction mechanism. As noted above, the fact that the reaction barriers for the two reaction paths differ by half an eV or more essentially guarantees a very high degree of bond selectivity in favor of the reaction channel with the lowest barrier. This is not a surprising result (it is essentially the basis of all of modern synthetic chemistry).

Instead, I feel the more interesting and impactful result is how prompt, electron stimulated reactions can lead to directed product recoil, and how the recoil process, when coupled with surface anisotropy, impacts the energy landscape between reactants and products. These ideas do appear in the paper, but in my reading of the paper, I found the connection between these ideas was obscured by the emphasis on bond selectivity.

Stated another way, it seems to me that the authors are using the prompt dissociation event to access different initial reaction paths/ trajectories for the two orientations of C-I bonds. The energy landscape along those two paths differs significantly due to the anisotropy of the surface, and this leads to the observed bond selectivity. This effect is illustrated very nicely in

Fig. 4. Perhaps connecting the discussion and conclusion more closely to this illustration would help convey this point to readers. [This is the reviewer's first point under E].

Second, the work emphasizes how important product binding can be in dictating reaction selectivity. Surface science has many examples of where differences in reagent binding (physisorption vs. chemisorption, or different chemisorption interactions) alter reactivity patterns, but this experiment provides a nice example of how product binding alone can influence reactivity.

So I'd encourage the authors to consider highlighting these other points in the paper.

Response to (E):

POINT 1: We have now re-examined the text. We find the first point in regard to the use of prompt dissociation event to access different reaction paths (a) is stated in the title (b) is stated next in the claim for novelty, page 2, para 2 (yellow highlighted), (c) then on p.6 in the second last paragraph of the Results (yellow highlighted) and finally (d) and (e) in the opening and closing sentences of the Discussion (p.7; yellow highlighted). Further discussion would be difficult to justify.

POINT 2: We were grateful for the suggestion, and have inserted the following sentence "This finding shows that the binding of the products to the surface, late in the course of reaction, can influence reaction barriers with resultant bond-selectivity"; see page 6, para 2, highlighted in red.

F) I don't see any need for additional experiments or calculations. See above for suggested improvements to the manuscript.

Also, on p. 4, the authors refer to the pDOS for the molecular as appearing in Fig. 3b. I believe it's in Fig. 2b.

Response to (F): We thank the reviewer for catching this misprint. We have now corrected it: on p.4 'Fig 3b' reads 'Fig 2b' (change highlighted in red).

G) References seem fine.

Response to (G): No question from Reviewer #1.

H) Clarity - see above comments above regarding the emphasis on bond selectivity.

Response to (H): Kindly refer to our response in (E).

Reviewer #2 (Remarks to the Author):

In the manuscript by Anggara et al., a selective bond dissociation of an adsorbate molecule is described that was investigated by using low-temperature STM. The dissociation of the C-I bond in meta-diiodobenzene is induced by the injection of a tunneling electron that is attached to the molecular state (LUMO). Most importantly, they found that the dissociation of one of the C-I bonds is strongly preferred compared to another because of the C-I bond orientation with respect to the anisotropic atomic arrangement of the Cu(110) surface. The experimental finding is supported by their theoretical calculations. The reported result is one of the clear examples that demonstrate the importance of atomic-scale environment on a chemical reaction at a metal surface. Particularly, the observed bond-selectivity is very difficult to identify with the other spectroscopic methods that observe a signal from molecular ensemble. I recommend the manuscript to be published in Nature communications.

Response: No question from Reviewer #2.

Questions from Reviewer #2:

1. Does the dissociation never occur at the opposite bias polarity? What happens when a hot hole is attached to the molecule?

Response to Q1: This study was limited to (hence entitled) electron-induced reaction; we did not look for hole-induced reaction.

2. How is the “average rate” and its error in Fig. 1a determined?

Response to Q2: The reviewer means Fig. 2a. This question was also asked by Reviewer #1 in his/her part (C). As noted above, a detailed response is to be found as the new insert in the Methods section (highlighted in red).

3. Is there any specific and general example that shows the importance of the selective bond dissociation presented in the manuscript? How is the knowledge obtained from single-molecule experiments useful to understand the process in a heterogeneous catalysis?

Response to Q3: The answer is now given in the red-highlighted text, page 7, where the closing sentence has been amended to read: “for example in chemical reactions at surfaces being subjected to generalized electron-radiation, in an extension of conventional heterogeneous catalysis.”

4. *Is the observed bond-selectivity unique for an electron-induced process? Is this also expected for a thermally-induced process?*

Response to Q4: The observed bond-selective pathway is indeed (as the Reviewer phrases it) “unique for an electron-induced process”. In response to the reviewer’s question the following sentences have been inserted on page 6 (red highlight) at the close of the Discussion:

“It should be noted that the bond-selectivity being reported is specific to the impulsive dynamics (see Fig.4), being a consequence of the directed-recoil of the atoms liberated in electron-induced reaction. The C-I bond-breaking could also in principle be obtained thermally, though this was not studied here. We have, however, calculated, *ab-initio*, the different energy-barriers to reaction for the individual breaking of the two C-I bonds by C-I extension along the alternative minimum-energy paths of the ground potential-energy surface. The computed barrier heights relevant to thermal reaction were found to be 0.3 eV for the previously favored ‘Along’ C-I, and 0.2 eV for the previously disfavored ‘Across’ C-I. Thermal reaction is therefore predicted to exhibit a modest bond-selectivity but in the opposite sense to that observed and modelled here for electron-induced reaction.”

END OF AUTHORS’ RESPONSES TO THE TWO REVIEWERS.

REVIEWERS' COMMENTS:

Reviewer #1 (Remarks to the Author):

Thank you for considering my suggestions to improve the manuscript. I support publication of the manuscript in its current form.

Reviewer #2 (Remarks to the Author):

The authors answered all of the concerns and questions from the reviewers and improved the manuscript. This should be accepted for publication.